# Synchronization of Complex Dynamical Networks with Stochastic Links Dynamics

**DOI:** 10.3390/e25101457

**Published:** 2023-10-17

**Authors:** Juanxia Zhao, Yinhe Wang, Peitao Gao, Shengping Li, Yi Peng

**Affiliations:** 1School of Automation, Guangdong University of Technology, Guangzhou 510006, China; 1112104030@mail2.gdut.edu.cn (J.Z.); yinhewang@gdut.edu.cn (Y.W.); pengyi@gdut.edu.cn (Y.P.); 2School of Electronics and Information, Guangdong Polytechnic Normal University, Guangzhou 510665, China; peitaogao@gpnu.edu.cn; 3MOE Key Laboratory of Intelligent Manufacturing, Shantou University, Shantou 515063, China

**Keywords:** stochastic complex dynamical network, mean square synchronization, dynamics of links, control strategy

## Abstract

The mean square synchronization problem of the complex dynamical network (CDN) with the stochastic link dynamics is investigated. In contrast to previous literature, the CDN considered in this paper can be viewed as consisting of two subsystems coupled to each other. One subsystem consists of all nodes, referred to as the nodes subsystem, and the other consists of all links, referred to as the network topology subsystem, where the weighted values can quantitatively reflect changes in the network’s topology. Based on the above understanding of CDN, two vector stochastic differential equations with Brownian motion are used to model the dynamic behaviors of nodes and links, respectively. The control strategy incorporates not only the controller in the nodes but also the coupling term in the links, through which the CDN is synchronized in the mean-square sense. Meanwhile, the dynamic stochastic signal is proposed in this paper, which is regarded as the auxiliary reference tracking target of links, such that the links can track the reference target asymptotically when synchronization occurs in nodes. This implies that the eventual topological structure of CDN is stochastic. Finally, a comparison simulation example confirms the superiority of the control strategy in this paper.

## 1. Introduction

Complex dynamical networks (CDNs) have received a lot of attention due to their wide application in transportation networks, telephone networks, internet networks, and many other real networks [1]. As a typical collective behavior exhibited in CDNs, synchronization has been seen as one of the most significant dynamical behaviors, and many interesting results have been reported, such as [2,3,4,5,6,7]. For example, in [7,8], the effect of the coupling strength between nodes and links on synchronization is investigated. It is shown that scintillating coupling enhances the synchronization of nodes in the network as well as how to determine the critical coupling strength.

From the perspective of graph theory, a typical CDN can be seen as the combination of nodes and links, where all the links show the layout of nodes and represent the network topology geometrically. This inspires us to consider all the links wholly as the dynamic subsystem, which is coupled with the other subsystem consisting of all the nodes. This suggests that a CDN can be thought of as a composite system with two subsystems, one is called the nodes subsystem (NS) and the other is the network topology subsystem (NTS), where the weighted values of links are viewed as the state variables of the NTS. According to the above view for the CDN, the NTS not only reflects the wholly dynamic change of network topology quantitatively but also affects the dynamic behaviors of NS via the coupling relationship between the NS and NTS [9,10,11]. In particular, the NTS can help the NS to achieve synchronization [9,10]. In other words, synchronization in the above literature is seen as the typical collective behavior of nodes with the link dynamics. However, the literature mentioned above ignores the effects of stochastic elements on the CDN.

It is worth noting that in real networks, the stochastic phenomena can be seen everywhere, which is often considered as the disturbance or noise acting on the network [12,13,14,15,16,17,18,19]. For example, in a neuronal network, each neuron is considered as a node, and the synapses between neurons are considered as links. From a neurophysiological point of view, biological neurons (nodes) are inherently random because the neural network receives the same stimulus repeatedly, but their responses are not the same [17]. The synapses (links) between neurons (nodes) also have randomness, which is caused by multiplicative noise at the synapses (links) [18,19]. Noise can enhance or weaken the transmission of neurotransmitters between neurons, so randomness plays an important role in biological neural networks. Similarly, in the communication transmission network, the links reflect the transmission between signals, which is often affected by network bandwidth, conduction medium, measurement noise, and other factors, which can result in the random loss or incomplete information. The above examples show that the noise has a crucial effect on both nodes and links. Therefore, it is better to consider both the effect of stochastic factors on nodes and links in CDN to reflect the essential properties of real systems as much as possible.

Therefore, controlling the CDN with stochastic disturbance and noise to achieve synchronization has become a hot issue in the existing literature [20,21,22,23,24,25,26,27,28]. For example, the problem of node synchronization control in dynamical networks with stochastic disturbance is discussed in [20]. A mean-square asymptotic synchronization criterion for stochastic complex networks with mixed time lags and multiple random perturbations is developed in [21]. In [22], the node dynamic equation with stochastic disturbance and time lag is established in the discrete system. Based on this, a controller is designed to make the CDN realize synchronization in the mean-square sense. However, the above results ignore the dynamics of NTS and certainly do not consider the influence of stochastic noise on NTS.

This paper, which was motivated by the above discussion, focuses on the impact of stochastic noise on the NS and NTS and proposes a control strategy that would allow all nodes to attain synchronization in the mean-square sense. In other words, two stochastic differential equations depict the dynamics of NS and NTS, where the NTS plays the auxiliary role in helping the NS achieve synchronization via the coupling relation between NS and NTS. It is worth noting that in this paper, the CDN consists of the NS and NTS, and the weights of the links are the state variable in NTS. However, in practical engineering applications, it is difficult to obtain accurate measurements of the weighted values of NTS due to technical constraints and measurement costs. That is to say, the state variable of the NTS is unavailable in the control strategy. In this case, if only the state variables of nodes can be available, how to synthesize the control strategy for the CDN with stochastic disturbance to achieve the above synchronization is worth discussing.

Motivated by the aforementioned discussions, this paper’s innovation focuses on three primary points:**(i)** Two stochastic differential equations are used to model the dynamics of nodes and links. In particular, the stochastic differential equation is used to model the dynamics of the links, which is rare in existing studies.**(ii)** The synchronization control method consists of not only the controller in the nodes but also the coupling term in the links. The stochastic complex dynamic network is synchronized in the mean-square sense under the action of these two components.**(iii)** The topology of the final network is stochastic and this result is unique in the existing literature. This is because when the nodes achieve synchronization, the links also track the stochastic auxiliary reference tracking target (ARTT). Compared with the existing literature, the key to the above innovation is that stochastic noise is introduced into the dynamic model of the links. This increases the analysis difficulty of CDN achieving the synchronization.

The remainder of the paper is structured as follows. The model for the dynamic changes of nodes and links involving the stochastic noise effect, and containing some required assumptions and lemma, is provided in Section 2. The control objective and the control strategy are provided in Section 3. To demonstrate the viability of the recommended control strategy, a numerical simulation is presented in Section 4. Finally, the conclusion is provided in Section 5.

**Notation** **1.** *$diag\{a{}_{1},a_{2},\cdots ,a_{n}\}$ represents a diagonal matrix with diagonal elements $a{}_{1},a_{2},\cdots ,a_{n}$; col{·} indicates the column vector; E{·} denotes the mathematical expectation operator relative to a given probability measure P; $∥\cdot ∥$ denotes the norm in Euclidean space of “·”;* ⊗ *represents the Kronecker product; $R^{n}$ represents n-dimensional Euclidean space; $R^{m\times n}$ represents m×n real matrices; $I_{N}$ denotes the identity matrix of the N order; Ones(N,1) represents a matrix of dimension N×1 with all elements of 1; rand(1) is a function that is used to generate a random number uniformly distributed in the interval [0, 1). $\left(\Omega ,F,{\left\{F_{t}\right\}}_{t\geq 0},P\right)$ indicates a complete probability space with a natural filtration ${\left\{F_{t}\right\}}_{t\geq 0}$ satisfying the usual conditions (i.e., the filtration contains all P-null sets and is right-continuous). We denote by ${L_{F_{0}}}^{p}\left(\left(-\infty ,0\right];R^{n}\right)$ the family of all $F_{0}$−measurable $C(\left(-\infty ,0\right];R^{n})$−valued random variables ζ={ζ(r):−∞<r≤0}, such that $\underset{-\infty <r\leq 0}{\sup}E\left\{{\left|\zeta (r)\right|}^{p}\right\}<\infty$.*

## 2. Model Description and Control Design

Consider a complex dynamical network (CDN) composed of *N* nodes with stochastic effects, where the dynamic equation of the *i*th node is
(1)$$dz_{i}\left(t\right)=\left\{Az_{i}+h_{i}\left(z_{i}\right)+\alpha \tilde{\Gamma }\Lambda \left(z\right)\xi_{i}+u_{i}\right\}dt+g\left(z_{i},\xi_{i},t\right)d\omega \left(t\right),i=1,2,\cdots ,N$$
where $z_{i}=z_{i}\left(t\right)=\mathrm{col}\left\{z_{i1},z_{i2},\cdots ,z_{in}\right\}\in R^{n}$ denotes the state vector of the *i*th node at time *t*; $z={\left[{z_{1}}^{T}\left(z_{1}\right),{z_{2}}^{T}\left(z_{2}\right),\cdots ,{z_{N}}^{T}\left(z_{N}\right)\right]}^{T}\in R^{nN}$; the real constant matrix $A\in R^{n\times n}$; the continuous nonlinear vector function $h_{i}\left(z_{i}\right)\in R^{n}$; α>0 indicates the coupling strength; the internal coupling matrix is defined as $\tilde{\Gamma }=diag\left\{a{}_{1},a_{2},\cdots ,a_{n}\right\}\in R^{n\times n}$, $a_{j}>0,j=1,2,\cdots ,n$; the internal coupling function $\Lambda \left(z\right)\in R^{n\times N}$. $\xi_{i}={\left(\xi_{i1},\xi_{i2},\cdots ,\xi_{iN}\right)}^{T}\in R^{N},i=1,2,\cdots ,N$ represents the outgoing link vector of the *i*th node [9], $\xi_{ij}$ denotes the weight of the link between the *i*th node and *j*th node, $u_{i}\in R^{n}$ is the control input of the *i*th node, $g\left(z_{i},\xi_{i},t\right)\in R^{n}$ is a vector function representing the intensity of random noise, ω(t) is a standard one-dimensional Brown motion defined on a complete probability space. $d\omega \left(t\right)\in R^{1}$ is interpreted as white noise, which is used to depict a class of “noise” or “disturbance” in dynamical systems and satisfies E{dω(t)}=0, $E\{d\omega^{2}\left(t\right)\}=dt$. In this paper, we investigate the system suffering from this “noise”, referred to as an $It\hat{o}$ stochastic system.

**Remark** **1.** 
*Inspired by [25,27], this paper considers Equation (Equation 1) as the dynamic equation of the ith node. In [27], the ith node’s dynamic equation is expressed as $dz_{i}\left(t\right)=\left\{h_{i}\left(z_{i}\right)+\overset{N}{\underset{j=1}{\alpha \sum }}\xi_{ij}\tilde{\Gamma }z_{j}\left(t\right)+u_{i}\right\}dt+g\left(z_{i},t\right)d\omega \left(t\right),i=1,2,\cdots ,N$. It is worth noting that the link $\xi_{ij}$ is taken as $\xi_{ij}>0$ (constant or time-varying values) when the ith node and jth node are linked together; otherwise, $\xi_{ij}=0$. The link $\xi_{ij}$ considered in this paper has dynamical behavior; that is, the dynamic change of link $\xi_{ij}$ is modeled by a differential equation, similar to the nodes. In addition, in [9], the ith node’s dynamic equation is $dz_{i}\left(t\right)=\left\{Az_{i}+h_{i}\left(z_{i}\right)+\alpha \tilde{\Gamma }\Lambda \left(z\right)\xi_{i}+u_{i}\right\}dt$, which is the equation obtained when the stochastic noise intensity $g(z_{i},\xi_{i},t)=0$ in this paper. Here, although the dynamic behavior of the links is taken into account, we ignore the effect of stochastic factors on network dynamics.*


Inspired by [9], we consider the following dynamical equation for the *i*th outgoing link vector $\xi_{i}$ in the CDN with stochastic efforts
(2)$$d\xi_{i}\left(t\right)=\left\{A_{2}\xi_{i}+\Phi_{2}\left(z,t\right)z_{i}\right\}dt+\bar{g}\left(z_{i},\xi_{i},t\right)d\omega \left(t\right)$$
where $A_{2}\in R^{N\times N}$ is the real constant matrix, $\Phi_{2}\left(z,t\right)\in R^{N\times n}$ is a function matrix representing the internal coupling relationship between nodes and links, $\bar{g}\left(z_{i},\xi_{i},t\right)\in R^{N}$ is the vector function representing random noise intensity.

**Remark** **2.** ***(i)** The connection relationship between the ith node and all other nodes is denoted by $\xi_{i}$, which is called the outgoing link vector of the ith node. If a matrix $\Xi ={\left[{\xi_{i}}_{j}\right]}_{N\times N}$ depicts the entire network’s topology, then $\xi_{i}$ is the ith column of matrix* Ξ*, which provides a clearer demonstration of each node $z_{i}$ connected to other nodes. Unlike the dynamic equation of links in [9], which is $d\xi_{i}\left(t\right)=\left\{A_{2}\xi_{i}+\Phi_{2}\left(z,t\right)z_{i}\right\}dt$, this paper considers the influence of stochastic factors on the state of links. **(ii)** In this paper, we consider two differential equations to model the dynamics of nodes and links separately. This modeling method is based on the perspective of this paper; that is, CDNs are coupled by nodes and links, and the dynamic change of either one of them will affect the other. At the same time, stochastic factors are taken into account, not only in the nodes but also in the links.**Setting $h\left(z\right)={\left[{h_{1}}^{T}\left(z_{1}\right),{h_{2}}^{T}\left(z_{2}\right),\cdots ,{h_{N}}^{T}\left(z_{N}\right)\right]}^{T}\in R^{nN}$, $u={\left[{u_{1}}^{T},{u_{2}}^{T},\cdots ,{u_{N}}^{T}\right]}^{T}\in R^{nN}$, $\xi =\xi \left(t\right)=\left[{\xi_{1}}^{T},{\xi_{2}}^{T},\cdots ,{\xi_{N}}^{T}\right]\in R^{N^{2}}$, $g\left(z,\xi ,t\right)=[g{\left(z_{1},\xi_{1},t\right)}^{T},g{\left(z_{2},\xi_{2},t\right)}^{T},\cdots ,$$g{\left(z_{N},\xi_{N},t\right)}^{T}{]}^{T}\in R^{nN}$, $\bar{g}\left(z,\xi ,t\right)={\left[\bar{g}{\left(z_{1},\xi_{1},t\right)}^{T},\bar{g}{\left(z_{2},\xi_{2},t\right)}^{T},\cdots ,\bar{g}{\left(z_{N},\xi_{N},t\right)}^{T}\right]}^{T}\in R^{N^{2}}$ and using the Kronecker product *⊗*, the above equations for the dynamics of the nodes (1) and links (2) can be integrated as follows:*(3)$$d\left(z\right)=\left\{\left(I_{N}⊗A\right)z+h\left(z\right)+\alpha \left[I_{N}⊗\Lambda \left(z\right)\right]\xi +u\right\}dt+g\left(z,\xi ,t\right)d\omega \left(t\right)$$(4)$$d\left(\xi \right)=\left\{\left(I_{N}⊗A_{2}\right)\xi +\left(I_{N}⊗\Phi_{2}\left(z,t\right)\right)z\right\}dt+\bar{g}\left(z,\xi ,t\right)d\omega \left(t\right)$$

**Remark** **3.** 
*Stochastic disturbance is a significant factor in causing network instability and poor performance in real-world applications [13]. Therefore, this modeling method, which considers stochastic perturbation in dynamic equations of both nodes and links, is closer to the real network.*

*Consider a given bounded differentiable reference signal $s^{*}\left(t\right)$ in the nodes, which is disturbed by the same noise intensity as nodes $z_{i}\left(t\right)$; that is*

(5)
$$d\left(s^{*}\right)=f^{*}\left(s^{*},\xi^{*},t\right)dt+g\left(s^{*},\xi^{*},t\right)d\omega \left(t\right)$$

*where $s^{*}\in R^{n}$, $f^{*}\left(s^{*},\xi^{*},t\right)\in R^{n}$ can be given arbitrarily, $g\left(s^{*},\xi^{*},t\right)\in R^{n}$, $\xi^{*}$ is the ARTT of the links. Let $z^{*}=\chi ⊗s^{*}(\chi =Ones\left(N,1\right)$), $f^{*}\left(z^{*},\xi^{*},t\right)=\chi ⊗f^{*}\left(s^{*},\xi^{*},t\right)\in R^{nN}$, $g\left(z^{*},\xi^{*},t\right)=\chi ⊗g\left(s^{*},\xi^{*},t\right)\in R^{nN}$; thus, Equation (Equation 5) can be rewritten as*

(6)
$$d\left(z^{*}\right)=f^{*}\left(z^{*},\xi^{*},t\right)dt+g\left(z^{*},\xi^{*},t\right)d\omega \left(t\right)$$


*Meanwhile, in order to assist the nodes in achieving asymptotic state synchronization, the ARTT $\xi^{*}$ of the links considered in this paper satisfies the following stochastic differential equation:*

(7)
$$d\left(\xi^{*}\right)=\left\{\left(I_{N}⊗A_{2}\right)\xi^{*}+\left(I_{N}⊗\Phi_{2}\left(z,t\right)\right)z^{*}\right\}dt+\bar{g}\left(z^{*},\xi^{*},t\right)d\omega \left(t\right)$$

*where $\xi^{*}\in R^{N^{2}}$, $\bar{g}\left(z^{*},\xi^{*},t\right)\in R^{N^{2}}$.*


**Remark** **4.** 
*Since the layout of the network topology plays a non-negligible role in realizing state synchronization of the nodes, the topology consisting of all the links considered in this paper is laid out according to the specified topology signal $\xi^{*}$, which can assist the nodes in achieving synchronization. That is, in the final time, when the nodes achieve state synchronization, the layout of the links is presented as the topology $\xi^{*}$.*


Two assumptions are given for Equations (1) and (2) to develop our main results in the future:

**Assumption** **1.** 
*For a continuous nonlinear vector function $h_{i}\left(t\right)$, there exists a known function $\tau_{i}\left(t\right)\geq 0$, such that $∥h_{i}\left(t\right)∥\leq \tau_{i}\left(t\right)$ holds, i=1,2,⋯N. In addition, matrices A and $A_{2}$, and noise intensity functions g(z,ξ,t) and $\bar{g}\left(z,\xi ,t\right)$ are known, and Λ(z) is known and bounded. It can be seen that $∥h(t)∥={\left[\sum_{i=1}^{N}{∥h_{i}\left(t\right)∥}^{2}\right]}^{\frac{1}{2}}\leq {\left[\sum_{i=1}^{N}{∥\tau_{i}\left(t\right)∥}^{2}\right]}^{\frac{1}{2}}\overset{\Delta }{\;=}\tau \left(t\right)$.*


**Assumption** **2.** 
*In the dynamic equation of the links, $A_{2}$ is Hurwitz.*

*From Assumption 2, we know that for any given positive definite matrix ${\bar{Q}}_{2}$, there is one and only one positive definite matrix ${\bar{P}}_{2}$ with suitable dimension, such that the following Lyapunov equation holds*

(8)
$${A_{2}}^{T}{\bar{P}}_{2}+{\bar{P}}_{2}A_{2}=-{\bar{Q}}_{2}$$


*Similarly, there exists a matrix K, such that $A_{1}=A-K$ is Hurwitz. Thus, similarly, the following Lyapunov equation can only be satisfied by one positive definite matrix ${\bar{P}}_{1}$ for each given positive definite matrix ${\bar{Q}}_{1}$.*

(9)
$${A_{1}}^{T}{\bar{P}}_{1}+{\bar{P}}_{1}A_{1}=-{\bar{Q}}_{1}$$


*Let $P_{1}=I_{N}⊗{\bar{P}}_{1}$, $P_{2}=I_{N}⊗{\bar{P}}_{2}$, $Q_{1}=I_{N}⊗{\bar{Q}}_{1}$, $Q_{2}=I_{N}⊗{\bar{Q}}_{2}$ and $P_{1}$, $P_{2}$, $Q_{1}$, and $Q_{2}$ are positive definite matrices. By applying the Kronecker product, the following equations can be obtained*

(10)
$${\left(I_{N}⊗A_{1}\right)}^{T}P_{1}+P_{1}\left(I_{N}⊗A_{1}\right)=-Q_{1}$$


(11)
$${\left(I_{N}⊗A_{2}\right)}^{T}P_{2}+P_{2}\left(I_{N}⊗A_{2}\right)=-Q_{2}$$



**Definition** **1** ([6]). *Think about the stochastic complex dynamical network described by (1) and (2) (or (3) and (4)), if $\lim_{t\to +\infty }E\left\{{∥z_{i}\left(t\right)-s^{*}\left(t\right)∥}^{2}\right\}=0$ holds, i=1,2,⋯,N, then the stochastic complex dynamical network is said to be asymptotically synchronized in the mean square.*

**Remark** **5.** 
*Based on the above symbols, it is clear that $\lim_{t\to +\infty }E\left\{{∥z_{i}\left(t\right)-s^{*}\left(t\right)∥}^{2}\right\}=0,i=1,2,\cdots ,N$ implies $\lim_{t\to +\infty }E\left\{{∥z\left(t\right)-z^{*}\left(t\right)∥}^{2}\right\}=0$.*


**Lemma** **1** ([28,29]). *Assuming f is a nonnegative function defined on [0,+∞), and it is Lebesgue-integrable and uniformly continuous on [0,+∞), then $\lim_{t\to +\infty }f\left(t\right)=0$.*

## 3. Design of Controller

We introduce the synchronization error vector $e_{i}\left(t\right)=z_{i}\left(t\right)-s^{*}\left(t\right)$, $e=e\left(t\right)=[e_{1}^{T},e_{2}^{T},\cdots ,$$e_{N}^{T}{]}^{T}$; we can see that the node synchronization error vector $e=e\left(t\right)=z\left(t\right)-z^{*}\left(t\right)$ and the tracking error vector $e_{\xi }=e_{\xi }\left(t\right)=\xi \left(t\right)-\xi^{*}\left(t\right)$ for the links.

**Control objective.** We consider the stochastic complex dynamical network composed of (1) and (2). For a given reference signal $z^{*}\left(t\right)$, we design the controller *u* in nodes and the coupling term $\Phi_{2}$ in links, such that the synchronization error $\lim_{t\to +\infty }E\left\{{∥e(t)∥}^{2}\right\}=\lim_{t\to +\infty }E\left\{{∥z\left(t\right)-z^{*}\left(t\right)∥}^{2}\right\}=0$ holds. That is, the stochastic complex dynamic network achieves asymptotic synchronization in the mean square.

The controller *u* for nodes in Equation (Equation 3) and the coupling term $\Phi_{2}\left(z,t\right)$ for links in Equation (Equation 4) are constructed as follows to fulfill the aforementioned control objective.
(12)$$u=-\left(I_{N}⊗K\right)e-\left(I_{N}⊗A\right)z^{*}-\alpha \left(I_{N}⊗\Lambda \left(z\right)\right)\xi^{*}+f^{*}\left(z^{*},\xi^{*},t\right)+v_{1}$$
(13)$$v_{1}=\left\{\begin{array}{c} -\frac{P_{1}e}{∥P_{1}e∥}\tau \left(z\right),e\neq 0\\ 0,e=0 \end{array}\right.$$
(14)$$\Phi_{2}\left(z,t\right)=-\alpha {P_{2}}^{-1}\Lambda {\left(z\right)}^{T}P_{1}$$

**Remark** **6.** 
*The control strategy for the aforementioned control objective is given by Equations (12)–(14). In the process of the control strategy designing, the information we use is the node state z, reference signals $z^{*}$ and $\xi^{*}$, and some known information in Equations (1) and (2). It should be noted that we cannot use the state information of the links since it is challenging to precisely obtain the links’ state. For example, in the winding system, the speed of the motors (nodes) can be easily measured by sensors, while the tension (links) between the motors is hard to measure correctly by the suitable sensors. u is called the controller of the nodes, which is composed of three parts. The first part $-(I_{N}⊗K)e$ is the error feedback term, where K is the gain matrix, which can be obtained by solving Equation (Equation 9), the second part $-\left(I_{N}⊗A\right)z^{*}-\alpha \left(I_{N}⊗\Lambda \left(z\right)\right)\xi^{*}+f^{*}\left(z^{*},\xi^{*},t\right)$ is the term related to the reference signals, where $z^{*}$, $\xi^{*}$ and $f^{*}$ contain information about the stochastic effect. This is different from the controllers designed in [9] that do not take the stochastic effect into account. The third part $v_{1}$ is called the robust term, which aims to overcome the uncertainty h(z) in Equation (Equation 3). In addition, we did not use stochastic information when designing the coupling term $\Phi_{2}$ in Equation (Equation 14).*

*Therefore, according to Equations (3) and (6), the dynamic equation of the nodes’ synchronization error can be derived:*

(15)
$$\begin{array}{cc} \; & de\left(t\right)=dz\left(t\right)-dz^{*}\left(t\right)\\ & =\left[\left(I_{N}⊗A\right)z+h\left(z\right)+\alpha \left(I_{N}⊗\Lambda \left(z\right)\right)\xi +u\right]dt+g\left(z,\xi ,t\right)d\omega \left(t\right)\\ & \;-[f^{*}\left(z^{*},\xi^{*},t\right)dt+g\left(z^{*},\xi^{*},t\right)d\omega \left(t\right)]\\ & =[\left(I_{N}⊗\left(A-K\right)\right)\left(z-z^{*}\right)+h\left(z\right)+\alpha \left(I_{N}⊗\Lambda \left(z\right)\right)\left(\xi -\xi^{*}\right)+\left(I_{N}⊗K\right)z\\ & \;+\left(I_{N}⊗\left(A-K\right)\right)z^{*}+\alpha \left(I_{N}⊗\Lambda \left(z\right)\right)\xi^{*}+u-f^{*}\left(z^{*},\xi^{*},t\right)]dt\\ & \;+\left[g\left(z,\xi ,t\right)-g\left(z^{*},\xi^{*},t\right)\right]d\omega \left(t\right)\\ & =[\left(I_{N}⊗A_{1}\right)e+h\left(z\right)+\alpha \left(I_{N}⊗\Lambda \left(z\right)\right)e_{\xi }+\left(I_{N}⊗K\right)z+\left(I_{N}⊗A_{1}\right)z^{*}\\ & \;+\alpha \left(I_{N}⊗\Lambda \left(z\right)\right)\xi^{*}+u-f^{*}\left(z^{*},\xi^{*},t\right)]dt+\left[g\left(z,\xi ,t\right)-g\left(z^{*},\xi^{*},t\right)\right]d\omega \left(t\right)\\ & =\left\{\left(I_{N}⊗A_{1}\right)e+\alpha \left(I_{N}⊗\Lambda \left(z\right)\right)e_{\xi }+h\left(z\right)+v_{1}\right\}dt+\left[g\left(z,\xi ,t\right)-g\left(z^{*},\xi^{*},t\right)\right]d\omega \left(t\right)\\ & =f_{1}dt+g_{1}d\omega \left(t\right) \end{array}$$


*Let $f_{1}=\left(I_{N}⊗A_{1}\right)e+\alpha \left(I_{N}⊗\Lambda \left(z\right)\right)e_{\xi }+h\left(z\right)+v_{1}$ and $g_{1}=g\left(z,\xi ,t\right)-g\left(z^{*},\xi^{*},t\right)$. Meanwhile, the dynamic equations of the tracking error of the links according to Equations (4) and (7) are derived as follows:*

(16)
$$\begin{array}{cc} \; & de_{\xi }\left(t\right)=d\xi \left(t\right)-d\xi^{*}\left(t\right)\\ & =\left[\left(I_{N}⊗A_{2}\right)\xi +\left(I_{N}⊗\Phi_{2}\left(z,t\right)\right)z\right]dt+\bar{g}\left(z,\xi ,t\right)d\omega \left(t\right)\\ & \;-[\left\{\left(I_{N}⊗A_{2}\right)\xi^{*}+\left(I_{N}⊗\Phi_{2}\left(z,t\right)\right)z^{*}\right\}dt+\bar{g}\left(z^{*},\xi^{*},t\right)d\omega \left(t\right)]\\ & =[\left(I_{N}⊗A_{2}\right)\left(\xi -\xi^{*}\right)+\left(I_{N}⊗\Phi_{2}\left(z,t\right)\right)\left(z-z^{*}\right)\\ & \;+\left(I_{N}⊗A_{2}\right)\xi^{*}+\left(I_{N}⊗\Phi_{2}\left(z,t\right)\right)z^{*}-\{\left(I_{N}⊗A_{2}\right)\xi^{*}\\ & \;+\left(I_{N}⊗\Phi_{2}\left(z,t\right)\right)z^{*}\}]dt+\left[\bar{g}\left(z,\xi ,t\right)-\bar{g}\left(z^{*},\xi^{*},t\right)\right]d\omega \left(t\right)\\ & =\left\{\left(I_{N}⊗A_{2}\right)e_{\xi }+\left(I_{N}⊗\Phi_{2}\left(z,t\right)\right)e\right\}dt+\left[\bar{g}\left(z,\xi ,t\right)-\bar{g}\left(z^{*},\xi^{*},t\right)\right]d\omega \left(t\right)\\ & =f_{2}dt+g_{2}d\omega \left(t\right) \end{array}$$


*Let $f_{2}=\left(I_{N}⊗A_{2}\right)e_{\xi }+\left(I_{N}⊗\Phi_{2}\left(z,t\right)\right)e$ and $g_{2}=\bar{g}\left(z,\xi ,t\right)-\bar{g}\left(z^{*},\xi^{*},t\right)$.*


**Assumption** **3** ([5,30]). *The noise intensity functions g and $\bar{g}$ satisfy the Lipschitz condition; that is, there are constants $\delta_{1}>0,{\bar{\delta }}_{1}>0,\delta_{2}>0,{\bar{\delta }}_{2}>0$, such that the following inequalities are satisfied:*
(17)$$\begin{array}{cc} \; & trace[{\left(g\left(z,\xi ,t\right)-g\left(z^{*},\xi^{*},t\right)\right)}^{T}\left(g\left(z,\xi ,t\right)-g\left(z^{*},\xi^{*},t\right)\right)]\\ & \;\leq \delta_{1}{∥z-z^{*}∥}^{2}+{\bar{\delta }}_{1}{∥\xi -\xi^{*}∥}^{2}=\delta_{1}{∥e∥}^{2}+{\bar{\delta }}_{1}{∥e_{\xi }∥}^{2} \end{array}$$*and*
(18)$$\begin{array}{cc} \; & trace[{\left(\bar{g}\left(z,\xi ,t\right)-\bar{g}\left(z^{*},\xi^{*},t\right)\right)}^{T}\left(\bar{g}\left(z,\xi ,t\right)-\bar{g}\left(z^{*},\xi^{*},t\right)\right)]\\ & \;\leq \delta_{2}{∥z-z^{*}∥}^{2}+{\bar{\delta }}_{2}{∥\xi -\xi^{*}∥}^{2}=\delta_{2}{∥e∥}^{2}+{\bar{\delta }}_{2}{∥e_{\xi }∥}^{2} \end{array}$$

**Theorem** **1.** 
*Consider the CDN with stochastic perturbations consisting of (1) and (2) (or (3) and (4)), and assume that Assumptions 1–3 and $\lambda_{\min}\left(Q_{1}\right)>\delta_{1}\lambda_{\max}\left(P_{1}\right)+\delta_{2}\lambda_{\max}\left(P_{2}\right)$, $\lambda_{\min}\left(Q_{2}\right)>{\bar{\delta }}_{1}\lambda_{\max}\left(P_{1}\right)+{\bar{\delta }}_{2}\lambda_{\max}\left(P_{2}\right)$ are satisfied, then by applying the control strategy to the CDN, the synchronization error $\lim_{t\to +\infty }E\left\{{∥z\left(t\right)-z^{*}\left(t\right)∥}^{2}\right\}=0$ holds; that is to say, the CDN is asymptotically synchronized in the mean square.*


**Proof of Theorem** **1.**Consider the positive definite function $V\left(t,e,e_{\xi }\right)=V_{1}\left(t,e\right)+V_{2}\left(t,e_{\xi }\right)=$$e^{T}\left(t\right)P_{1}e\left(t\right)+e_{\xi }^{T}\left(t\right)P_{2}e_{\xi }\left(t\right)$, where $V_{1}=V_{1}\left(t,e\right)=e{\left(t\right)}^{T}P_{1}e\left(t\right)$ and $V_{2}=V_{2}\left(t,e_{\xi }\right)=e_{\xi }^{T}\left(t\right)P_{2}e_{\xi }\left(t\right)$. Let $\beta =\lambda_{\min}\left(Q_{1}\right)-\delta_{1}\lambda_{\max}\left(P_{1}\right)-\delta_{2}\lambda_{\max}\left(P_{2}\right)$ and $\gamma =\lambda_{\min}\left(Q_{2}\right)-{\bar{\delta }}_{1}\lambda_{\max}\left(P_{1}\right)-{\bar{\delta }}_{2}\lambda_{\max}\left(P_{2}\right)$. The derivatives of $V(t,e,e_{\xi })$ is $d\left(V\right)=d\left(V_{1}\right)+d\left(V_{2}\right)$. Using the $It\hat{o}$ differential formula
(19)$$d\left(V_{1}\right)=LV_{1}\left(t,e\right)dt+V_{{}_{1e}}^{T}\left(t,e\right)g_{1}d\omega \left(t\right)$$
(20)$$d\left(V_{2}\right)=LV_{2}\left(t,e_{\xi }\right)dt+V_{{}_{2e_{\xi }}}^{T}\left(t,e_{\xi }\right)g_{2}d\omega \left(t\right)$$
where
(21)$$\begin{array}{cc} \; & LV_{1}=\frac{\partial V_{1}}{\partial t}+{\left(\frac{\partial V_{1}}{\partial e}\right)}^{T}f_{1}+\frac{1}{2}trace\left[{g_{1}}^{T}{V_{1}}_{ee}g_{1}\right]\\ & =2\left(e^{T}P_{1}\right)f_{1}+trace\left[{g_{1}}^{T}P_{1}g_{1}\right]\\ & =2e^{T}P_{1}\left[\left(I_{N}⊗A_{1}\right)e+\alpha \left(I_{N}⊗\Lambda \left(z\right)\right)e_{\xi }+h\left(z\right)+v_{1}\right]+trace\left[{g_{1}}^{T}P_{1}g_{1}\right] \end{array}$$
and
(22)$$\begin{array}{cc} \; & LV_{2}=\frac{\partial V_{2}}{\partial t}+{\left(\frac{\partial V_{2}}{\partial e_{\xi }}\right)}^{T}f_{2}+\frac{1}{2}trace\left[{g_{2}}^{T}{V_{2}}_{e_{\xi }e_{\xi }}g_{2}\right]\\ & =2\left({e_{\xi }}^{T}P_{2}\right)f_{2}+trace\left[{\bar{g}}^{T}P_{2}\bar{g}\right]\\ & =2\left({e_{\xi }}^{T}P_{2}\right)\left[\left(I_{N}⊗A_{2}\right)e_{\xi }+\left(I_{N}⊗\Phi_{2}\left(z,t\right)\right)e\right]+trace\left[{g_{2}}^{T}P_{2}g_{2}\right] \end{array}$$According to Assumption 3, we can deduce
(23)$$\begin{array}{cc} \; & LV=LV_{1}+LV_{2}\\ & =2e^{T}P_{1}\left[\left(I_{N}⊗A_{1}\right)e+\alpha \left(I_{N}⊗\Lambda \left(z\right)\right)e_{\xi }+h\left(z\right)+v_{1}\right]\\ & \;+2\left({e_{\xi }}^{T}P_{2}\right)\left[\left(I_{N}⊗A_{2}\right)e_{\xi }+\left(I_{N}⊗\Phi_{2}\left(z,t\right)\right)e\right]+trace\left[{g_{1}}^{T}P_{1}g_{1}+{g_{2}}^{T}P_{2}g_{2}\right]\\ & =2e^{T}P_{1}\left(I_{N}⊗A_{1}\right)e+2e^{T}P_{1}\left[\alpha \left(I_{N}⊗\Lambda \left(z\right)\right)e_{\xi }+h\left(z\right)+v_{1}\right]\\ & \;+2{e_{\xi }}^{T}P_{2}\left[\left(I_{N}⊗A_{2}\right)e_{\xi }\right]+2{e_{\xi }}^{T}P_{2}\left[\left(I_{N}⊗\Phi_{2}\left(z,t\right)\right)e\right]+trace\left[{g_{1}}^{T}P_{1}g_{1}+{g_{2}}^{T}P_{2}g_{2}\right]\\ & =2e^{T}P_{1}\left(I_{N}⊗A_{1}\right)e+2{e_{\xi }}^{T}P_{2}\left[\left(I_{N}⊗A_{2}\right)e_{\xi }\right]\\ & \;+2e^{T}P_{1}\left[\alpha \left(I_{N}⊗\Lambda \left(z\right)\right)e_{\xi }+h\left(z\right)+v_{1}\right]+2{e_{\xi }}^{T}P_{2}\left[\left(I_{N}⊗\Phi_{2}\left(z,t\right)\right)e\right]\\ & \;+trace[{g_{1}}^{T}P_{1}g_{1}+{g_{2}}^{T}P_{2}g_{2}]\\ & =e^{T}\left(P_{1}\left(I_{N}⊗A_{1}\right)+{\left(I_{N}⊗A_{1}\right)}^{T}P_{1}\right)e+{e_{\xi }}^{T}\left(P_{2}\left(I_{N}⊗A_{2}\right)+{\left(I_{N}⊗A_{2}\right)}^{T}P_{2}\right)e_{\xi }\\ & \;+2e^{T}P_{1}\left(h\left(z\right)+v_{1}\right)+trace\left[{g_{1}}^{T}P_{1}g_{1}+{g_{2}}^{T}P_{2}g_{2}\right]\\ & \leq -e^{T}\left(Q_{1}\right)e-{e_{\xi }}^{T}\left(Q_{2}\right)e_{\xi }+2e^{T}P_{1}\left(h\left(z\right)+\bar{v}\right)\\ & \;+\lambda_{\max}\left(P_{1}\right)trace\left[{g_{1}}^{T}g_{1}\right]+\lambda_{\max}\left(P_{2}\right)trace\left[{g_{2}}^{T}g_{2}\right]\\ & \leq -e^{T}\left(Q_{1}\right)e-{e_{\xi }}^{T}\left(Q_{2}\right)e_{\xi }+\lambda_{\max}\left(P_{1}\right)\delta_{1}\left({∥e∥}^{2}+{∥e_{\xi }∥}^{2}\right)+\lambda_{\max}\left(P_{2}\right)\delta_{2}\left({∥e∥}^{2}+{∥e_{\xi }∥}^{2}\right)\\ & \leq -\lambda_{\min}\left(Q_{1}\right){∥e∥}^{2}-\lambda_{\min}\left(Q_{2}\right){∥e_{\xi }∥}^{2}+\lambda_{\max}\left(P_{1}\right)(\delta_{1}{∥e∥}^{2}+{\bar{\delta }}_{1}{∥e_{\xi }∥}^{2}\\ & \;+\lambda_{\max}\left(P_{2}\right)\left(\delta_{2}{∥e∥}^{2}+{\bar{\delta }}_{1}{∥e_{\xi }∥}^{2}\right)\\ & =-\left(\lambda_{\min}\left(Q_{1}\right)-\delta_{1}\lambda_{\max}\left(P_{1}\right)-\delta_{2}\lambda_{\max}\left(P_{2}\right)\right){∥e∥}^{2}\\ & \;-\left(\lambda_{\min}\left(Q_{2}\right)-{\bar{\delta }}_{1}\lambda_{\max}\left(P_{1}\right)-{\bar{\delta }}_{2}\lambda_{\max}\left(P_{2}\right)\right){∥e_{\xi }∥}^{2}\\ & =-\beta {∥e∥}^{2}-\gamma {∥e_{\xi }∥}^{2}\\ & =-\beta {∥z\left(t\right)-z^{*}\left(t\right)∥}^{2}-\gamma {∥\xi \left(t\right)-\xi^{*}\left(t\right)∥}^{2} \end{array}$$
where β>0 and γ>0, thus $E\{L(V)\}\leq -\beta E\left\{{∥z\left(t\right)-z^{*}\left(t\right)∥}^{2}\right\}-\gamma E\left\{{∥\xi \left(t\right)-\xi^{*}\left(t\right)∥}^{2}\right\}$. For any t≥0, by using Equation (Equation 23), we have
(24)$$\begin{array}{cc} \; & E\{V\left(t\right)\}-E\{V\left(0\right)\}=E\{V_{1}\left(t\right)-V_{1}\left(0\right)\}+E\{V_{2}\left(t\right)-V_{2}\left(0\right)\}=\int_{0}^{t}E\{LV(s)\}ds\\ & \leq -\beta \int_{0}^{t}E\{{∥z\left(s\right)-z^{*}\left(s\right)∥}^{2}\}ds-\gamma \int_{0}^{t}E\{{∥\xi \left(s\right)-\xi^{*}\left(s\right)∥}^{2}\}ds \end{array}$$This implies that
(25)$$\int_{0}^{t}E\{{∥z\left(s\right)-z^{*}\left(s\right)∥}^{2}\}ds\leq \frac{1}{\beta }E\left\{V_{1}\left(0\right)\right\}-\frac{1}{\beta }E\left\{V_{1}\left(t\right)\right\}\leq \frac{1}{\beta }E\left\{V_{1}\left(0\right)\right\}$$
(26)$$\int_{0}^{t}E\{{∥\xi \left(s\right)-\xi^{*}\left(s\right)∥}^{2}\}ds\leq \frac{1}{\gamma }E\left\{V_{2}\left(0\right)\right\}-\frac{1}{\gamma }E\left\{V_{2}\left(t\right)\right\}\leq \frac{1}{\gamma }E\left\{V_{2}\left(0\right)\right\}$$
Thus $\int_{0}^{t}E\{{∥z\left(s\right)-z^{*}\left(s\right)∥}^{2}\}ds<+\infty$ and $\int_{0}^{t}E\{{∥\xi \left(s\right)-\xi^{*}\left(s\right)∥}^{2}\}ds<+\infty$.Furthermore, obtaining $E\{{∥z\left(t\right)-z^{*}\left(t\right)∥}^{2}\}$ and $E\{{∥\xi \left(t\right)-\xi^{*}\left(t\right)∥}^{2}\}$ as uniformly continuous on [0,+∞) is not difficult. Then, according to Lemma 1, we can obtain $\lim_{t\to +\infty }E\left\{{∥z\left(t\right)-z^{*}\left(t\right)∥}^{2}\right\}=0$ and $\lim_{t\to +\infty }E\left\{{∥\xi \left(t\right)-\xi^{*}\left(t\right)∥}^{2}\right\}=0$, which means that the CDN with stochastic perturbations is asymptotically synchronized in the mean square. This completes the proof of Theorem 1. □

**Remark** **7.** 
*$\lim_{t\to +\infty }E\left\{{∥z\left(t\right)-z^{*}\left(t\right)∥}^{2}\right\}=0$ and $\lim_{t\to +\infty }E\left\{{∥\xi \left(t\right)-\xi^{*}\left(t\right)∥}^{2}\right\}=0$ imply that when the nodes achieve synchronization, the NT tracks the given reference signal; that is, the NT is laid out according to the $\xi^{*}\left(t\right)$ and this layout is stochastic. This is novel in the existing literature.*


## 4. Simulation Example

Considering the CDN composed of the *N* underactuated surface ship, the kinematics equation of each ship is expressed as follows [31]:(27)$${\dot{s}}_{i}=\frac{d_{1}}{m_{1}}s_{i}+\frac{m_{2}}{m_{1}}w_{i}v_{i}+\frac{1}{m_{1}}\tau_{s_{i}}$$
(28)$${\dot{v}}_{i}=\frac{d_{3}}{m_{3}}v_{i}+\frac{m_{1}-m_{2}}{m_{3}}s_{i}w_{i}+\frac{1}{m_{3}}\tau_{v_{i}}$$
(29)$${\dot{w}}_{i}=\frac{d_{2}}{m_{2}}w_{i}+\frac{m_{1}}{m_{2}}s_{i}v_{i}$$
where $s_{i},v_{i},w_{i}$ denote the speed of surge, yaw, and sway of the *i*th ship. The control inputs are the surge force $\tau_{s_{i}}$ and the yaw moment $\tau_{v_{i}}$; parameters $m_{i}$, $d_{i}$, i=1,2,3 are positive constants, which denote the ship’s inertia and damping.

In this paper, we consider the synchronization of the surge speed and the yaw speed, and we do not require the sway speed (it is sufficient to keep bounded), so we define the state variable $z_{i}=\left[\begin{array}{c} s_{i}\\ v_{i} \end{array}\right]$. Equations (27) and (28) can be written as
(30)$${\dot{z}}_{i}=Az_{i}+h_{i}\left(z,t\right)+u_{i}$$
where $A=\left[\begin{array}{cc} \frac{-d_{1}}{m_{1}} & 0\\ 0 & \frac{-d_{3}}{m_{3}} \end{array}\right]$, $h_{i}\left(z,t\right)=\left[\begin{array}{c} \frac{m_{2}}{m_{1}}w_{i}v_{i}\\ \frac{m_{1}-m_{2}}{m_{3}}s_{i}w_{i} \end{array}\right]$, $u_{i}=\left[\begin{array}{cc} \frac{1}{m_{1}} & 0\\ 0 & \frac{1}{m_{3}} \end{array}\right]\left[\begin{array}{c} \tau_{u}\\ \tau_{r} \end{array}\right]$.

Inspired by [9], we consider the given communication protocol $\sum_{j=1}^{N}\xi_{ji}\tilde{\Gamma }\Lambda \left(z\right)$, where $\xi_{ji}$ represents the communication strength between the *j*th ship and the *i*th ship, and its dynamic equation is determined by Equation (Equation 2). In addition, considering the influence of stochastic factors as in [21], Equation (Equation 30) can be modified as follows:(31)$$dz_{i}=\left\{Az_{i}+h_{i}\left(t\right)+\alpha \sum_{j=1}^{N}\xi_{ji}\tilde{\Gamma }\Lambda \left(z\right)+u_{i}\right\}dt+g\left(z_{i},\xi_{i},t\right)d\omega \left(t\right)$$
Furthermore, the links’ dynamic equation ξ is selected as Equation (Equation 4). The steps for simulation in Matlab are as follows (*n* = 2, *N* = 10):

**Step 1** Determine the initial state of the nodes z(0) and the links ξ(0), respectively; $z\left(0\right)=\left[z_{1}{\left(0\right)}^{T},z_{2}{\left(0\right)}^{T},\cdots ,z_{N}{\left(0\right)}^{T}\right]\in R^{20}$, $z_{i}\left(0\right)=rand\left(2,1\right)$, i=1,2,⋯,N, ξ(0)=rand(100,1).

**Step 2** Determine the matrix functions *A*, $h_{i}\left(t\right)$, $\tilde{\Gamma }$, Λ(z), $g(z_{i},\xi_{i},t)$ and α in Equation (Equation 31). According to the parameters $m_{1}=0.5160$ kg, $m_{2}=14.4300$ kg, $m_{3}=0.5160$ kg, $d_{1}=12.1800$ kg/s, $d_{2}=207.8720$ kg/s, $d_{3}=0.4512$ kg/s, $w_{i}=\left(1.5+0.1\sin\left(it\right)\right)m/s$ of the ship in [31], *A*, $h_{i}\left(t\right)$ can be determined. In addition, we select $\tilde{\Gamma }=diag\left\{a_{1},a_{2}\right\}$, $\Lambda \left(z\right)=col\{\cos\left(z_{i1}\right),\cos\left(z_{i2}\right)\}$, $g=0.1col\{e_{i1},e_{i2}\}$, $\tau \left(z\right)=\sqrt{\sum_{i=1}^{N}\left[{\left(\frac{m_{2}}{m_{1}}w_{i}v_{i}\right)}^{2}+{\left(\frac{m_{1}-m_{2}}{m_{3}}s_{i}w_{i}\right)}^{2}\right]}$. In order to avoid chance in parameter selection, $a_{i}$ and α are chosen as $a_{1}=rand\left(1\right)$, $a_{2}=rand\left(1\right)$, α=rand(1). Let m$\left(z_{i},t\right)=Az_{i}\left(t\right)+h_{i}\left(z_{i}\right)+\alpha \tilde{\Gamma }\Lambda \left(z\right)\xi_{i}+u_{i}$ in Step 4.

**Step 3** Determine the matrix functions $A_{2}$ and $\bar{g}\left(z,\xi ,t\right)$ in Equation (Equation 4). According to Assumption 2, $A_{2}$ is Hurwitz. Thus, let $A_{2}=\sigma B\left[\begin{array}{cc} b_{1} & 0\\ 0 & b_{2} \end{array}\right]B^{-1}$, where *B* is a randomly generated invertible matrix of the *N* order, and $b_{i}=-rand\left(1\right)$, σ is an adjustable parameter and is chosen as σ=0.5, $\bar{g}=0.1e_{\xi }$.

**Step 4** Solving Equation (Equation 31), by using finite difference methods:

T = 2;

N = 2000;

dt = T/N;

for i = 1:N

t(i + 1) = t(i) + dt;

dω= sqrt(dt) * randn();

z(i + 1) = z(i) + m(z(i), t(i)) * dt + g(z(i), t(i)) * dω;

end

**Step 5** Give the reference signal of the nodes $s^{*}$ and choose $f^{*}\left(s^{*},\xi^{*},t\right)={\left[1,1\right]}^{T}$. At the same time, the auxiliary reference tracking target (ARTT) of the links is chosen as Equation (Equation 7).

**Step 6** Substitute the above parameters and matrices into the synthesized control strategy (12)–(14), which ensures the stochastic complex dynamical network is asymptotically synchronized in the mean square sense.

In order to highlight the benefits of the control strategy designed in this paper, the node synchronization error is compared with that when using the controller in [9]. In addition, the norm $∥e(t)∥=\sqrt{\sum_{i=1}^{N}{∥e_{i}\left(t\right)∥}^{2}}$ of errors is adopted in the comparison simulation.

From the simulation results in Figure 1, Figure 2, Figure 3, Figure 4 and Figure 5, the following observations can be drawn.

**(i)** From Figure 1 and Figure 2, it can be seen that without the control strategy, neither the synchronization error of nodes nor the tracking error of links tends to zero, while both errors tend to zero after the control strategy is applied, which means that the nodes achieve synchronization and the links track the given reference signals with the control strategy proposed in this paper.

**(ii)** From Figure 3 and Figure 4, it is evident that the reference signals of the nodes and links are bounded. At the same time, the state curves of nodes and links with a control strategy tend to be the same as their reference signals. In particular, when the nodes achieve synchronization, the network topology also tracks the given stochastic reference signal $\xi^{*}\left(t\right)$. That is, the final network layout is stochastic, which is rare in the existing results.

**(iii)**Figure 5 uses the synchronization error norm to compare the effectiveness of the controller in [9] and the controller proposed in this paper, where the controller in [9] does not consider stochastic factors, while the controller proposed in this paper contains stochastic information. It can be observed that the synchronization error norm tends to approach zero when using the controller proposed in this paper, while the controller in [9] does not. This indicates that the controller containing stochastic information designed in this paper is more suitable to realize node synchronization than the controller in [9].

## 5. Conclusions

In this paper, two vector differential equations are used for modeling the dynamics of nodes and links, accounting for the influence of stochastic factors. Unlike existing studies, we also consider the influence of stochastic factors in the links. The control strategy designed in this paper includes nodes controllers and coupling term in the links, and the combined effects of the two parts make the nodes realize synchronization. In addition, when the nodes reach synchronization, the links are laid out according to the specified reference signals, and this layout is stochastic. Inspired by the synchronization of nodes, it is interesting to consider the synchronization of links in CDNs with stochastic perturbations.

## Figures and Tables

**Figure 1 entropy-25-01457-f001:**
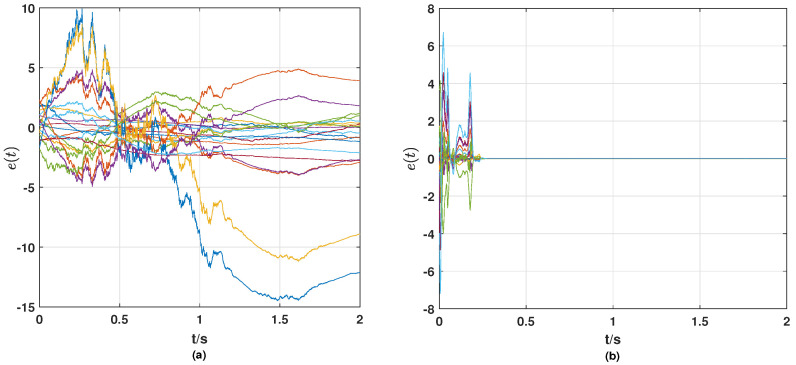
(**a**) Synchronization error e(t) of surge and yaw speeds (synchronization error of the nodes) of ships without control strategy; (**b**) synchronization error e(t) of surge and yaw speeds (synchronization error of the nodes) of ships with a control strategy.

**Figure 2 entropy-25-01457-f002:**
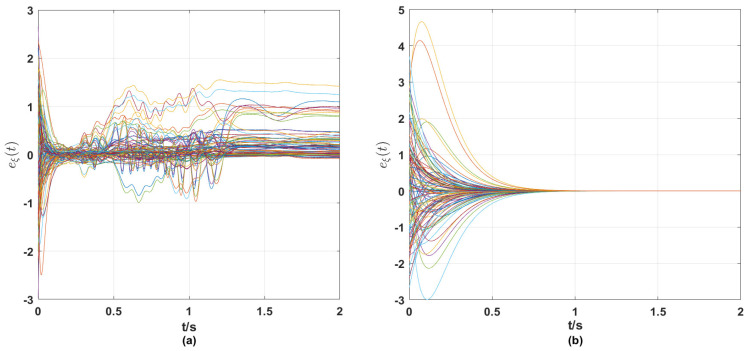
(**a**) Tracking error $e_{\xi }\left(t\right)$ of communication strength (tracking error of the links) between ships without control strategy; (**b**) tracking error $e_{\xi }\left(t\right)$ of communication strength (tracking error of the links) between ships with a control strategy.

**Figure 3 entropy-25-01457-f003:**
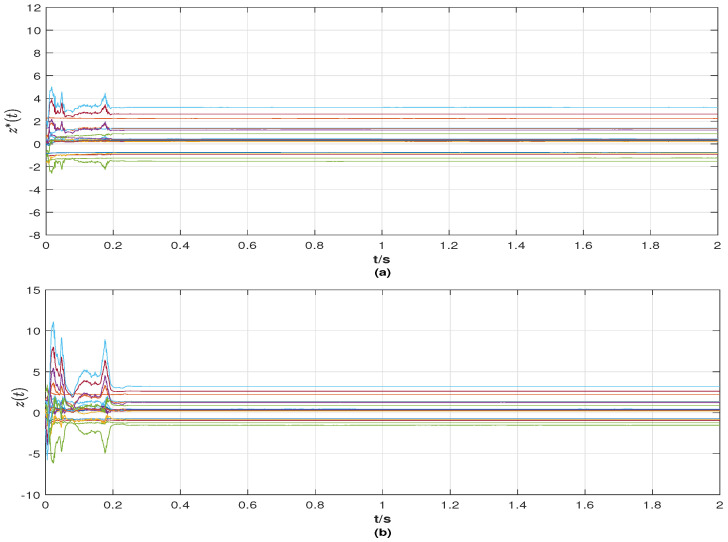
(**a**) Reference signals $z^{*}\left(t\right)$ of surge and yaw speeds (reference signals of nodes) of ships; (**b**) the state z(t) of surge and yaw speeds (state of nodes) of ships with a control strategy.

**Figure 4 entropy-25-01457-f004:**
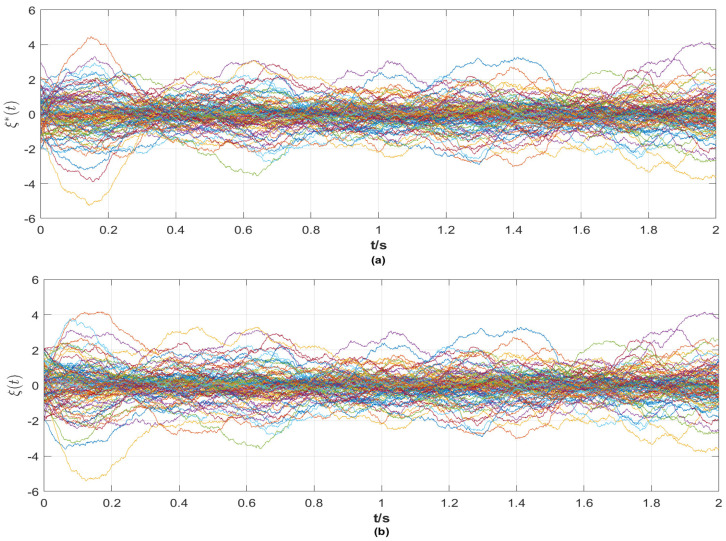
(**a**) The reference signals $\xi^{*}\left(t\right)$ of the communication strength (reference signals of links) between ships; (**b**) the state ξ(t) of the communication strength (state of links) between ships with a control strategy.

**Figure 5 entropy-25-01457-f005:**
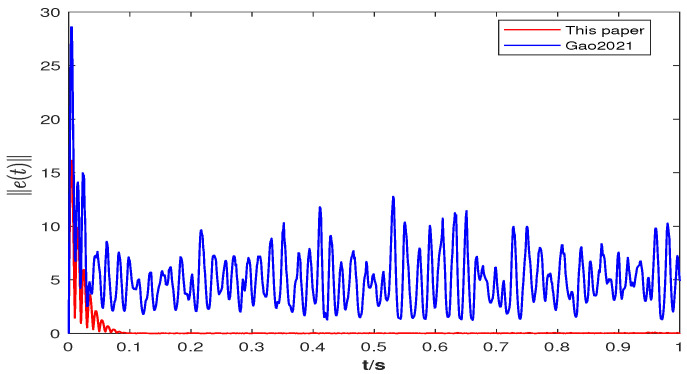
The synchronization error norm $∥e(t)∥$ of surge and yaw speeds (synchronization error norm of the nodes) of ships with a control strategy in Gao2021 [9] and this paper.

## Data Availability

Not applicable.

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
