# Peer review of "Synchronization of Complex Dynamical Networks with Stochastic Links Dynamics"

_entropy, 2023, doi:10.3390/e25101457_

Round 1

Reviewer 1 Report

The authors propose a control strategy for the mean square synchronization problem of the complex dynamical network (CDN) with the influence of stochastic factors.

The work is mathematically rigorous and in my opinion deserves publication.

Author Response

Thank you for your careful reading of our manuscript and your endorsement of this work.

Reviewer 2 Report

In "Synchronization of complex dynamical networks with stochastic links dynamics" the authors study a relevant problem with many potential applications and further research, especially for better understanding synchronization on various types of complex networks with changing links, either stochastic or aim-specific.

But similar analysis have been presented before, and it is not clear how groundbreaking and relevant this is for the readers of Entropy. If a revision is granted, the following comments should be taken into account with care and love to detail.

1) The authors should please state more clearly how their approach is related to existing approaches and how it advances the field in terms of ensuring synchronization of complex dynamical networks. How does this approach relevantly promote the state of the art, and how relevant is this for systems that are already now handled well by existing approaches? As it stands, I am not convinced in terms of novelty. This is mainly a modification of many similar approaches for synchronization in time-varying complex networks and a good justification should be provided for publication in Entropy.

2) It would also improve the paper if the figure captions would be made more self-contained. In addition to what is shown for which parameter values, one should also write a sentence or two saying what is the main message of each figure.

3) More importantly, the presentation of the results is very abstract, with too little guidance of the reader through the many algorithms and mathematical details. Since Entropy is not a purely computer science or applied mathematics journal, such style will likely not appeal and not be understandable to the majority of the readers. The authors must improve the clarity of the writing.

4) The introduction is missing several closely related references where synchronization in networks has been studied before, namely: Explosive synchronization dependence on initial conditions: The minimal Kuramoto model, Atiyeh Bayani, et al., Chaos, Solitons & Fractals 169, 113243 (2023) and Blinking coupling enhances network synchronization, Fatemeh Parastesh, et al., Phys. Rev. E 105, 054304 (2022). The introduction should be made more comprehensive and also related to more recent research in terms of reviews and synchronization.

5) Also, it would be very useful if the authors would make their source code available as supplementary material. This would promote the usage of the proposed approach and allow also others to take advantage of this research, and also to allow them to reproduce the results.

6) Some references contain errors and inconsistent formatting. It is difficult to give credit to research if even such elementary aspects of the work are not error free. The references should be made error free and formatted in agreement with the journal guidelines.

7) And finally, the description of the highlights should be improved for clarity and impact. And the abstract is also not particularly informative as to the new results of this research. There are also too many abbreviations, already in the abstract.

If a revision is granted, I will be happy to review the manuscript again.

Reviewer 3 Report

The manuscript presents an interestingly posed question of the role of synchronization of complex neural networks using noise. However, there are a few points of confusion probably resulting from the way the issue is presented.

(1) The authors use the assumption of the presence of Gaussian noise, about which they write imprecisely "is described by the Brownian motion, which is seen as white noise." One can only guess that the stochastic equations are to be used to model Wiener processes. A more formal definition does not appear until page 3 on line 119. In particular, stochastic equations e.g. Eq. 1 contain a random expression dw (d \n) without explanation.

(2) why d \omega in all stochastic equtions has no index? The formulas in this form suggest the same random numbers for each node and for both stochastic systems, NS and NTS.

(3) For a full understanding of the model, details of the computer simulation of Eq. 31, including the finite difference form of Eq. 31 used in the simulation. The authors only mentioned that they used Matlab.

(4)It is necessary to justify both the choice of the Wiener process and the choice of the uniform distribution (?) for random variables a1, a2, \alpha (Page 9, line 220). They have no indices in Eq.31. Should this be the case? In addition, the authors wrote, ” rand(1) means to generate a random number between 0 and 1". This definition is mathematically inaccurate.

(5) Page 1, Line 21: why ”Internet networks” - why with a capital letter?

In summary, the computer simulation results suggest the correctness of the stochastic equations used but the presentation of the model needs refinement. The presentation is not clear.

Round 2

Reviewer 2 Report

The authors have revised their manuscript comprehensively and with love to detail. I warmly recommend publication in present form.

Reviewer 3 Report

Basically, the authors responded to my concerns. The manuscript may be accepted for publication.